# The Impact of Storage Temperature on the Development of Microbial Communities on the Surface of Blueberry Fruit

**DOI:** 10.3390/foods12081611

**Published:** 2023-04-11

**Authors:** Junying Wang, Chong Shi, Donglu Fang, Jilu Che, Wenlong Wu, Lianfei Lyu, Weilin Li

**Affiliations:** 1Co-Innovation Center for Sustainable Forestry in Southern China, Forestry College, Nanjing Forestry University, Nanjing 210037, China; 2College of Food Engineering, Anhui Science and Technology University, Chuzhou 233100, China; 3Key Laboratory for the Research and Utilization of Plant Resources, Institute of Botany, Jiangsu Province and Chinese Academy of Sciences, Nanjing 210014, China

**Keywords:** blueberry, high-throughput sequencing, microbiome, biodiversity, temperature

## Abstract

Microbial contamination is one of the main reasons for the quality deterioration of postharvest blueberries during storage. In this study, we investigated the surface microbiota of blueberry fruits stored at different temperatures via high-throughput sequencing of the 16S and ITS rRNA genes. The results indicated that the α-diversity of the microbial communities in samples stored at 4 °C was much higher than that in samples stored at 25 °C. The composition of the bacterial and fungal communities on the surface of the blueberry fruits varied at different storage temperatures. Ascomycota, Basidiomycota, Anthophyta, Chlorophyta, Proteobacteria, and Cyanobacteria were the most abundant phyla in the bacterial community. Furthermore, five preservation quality indices were measured, and the influence on the α-diversity of the bacterial community was found to be significantly weaker than that of the fungal community. Based on the prediction of the bacterial flora function, the change in blueberry quality during storage was closely related to its surface microbial effect. This study provides a theoretical basis for an understanding of the microbiota on the surface of blueberry fruits to cause fruit spoilage, and the development of a targeted inhibition technology to preserve blueberry fruits under different storage and transportation environments.

## 1. Introduction

Blueberries (*Vaccinium* spp.) are a very attractive fruit mainly for their nutrition and health function [1]. Blueberries are considered a health-promoting fruit because of their high flavonoid content and antioxidant activity [2]. The blueberry planting areas and production have increased according to global blueberry industry data released by the International Blueberry Association (IBO) in 2021. Fresh blueberries are prone to quality deterioration, but their nutritional and economic value are largely determined by their preservation quality during the postharvest period [3]. The major causes of spoilage for blueberries are fungal decay and physiological changes [4]. The postharvest quality changes are closely related to microbial infestations, such as gray mold [5]. The storage temperature is usually considered an important factor in delaying the postharvest decay of fruits [6,7]. Cold storage was designed to control the postharvest respiration of fruits and inhibit the growth of microorganisms on the fruit surface, but the cold chain-system in China is not perfect. Since controlling microorganisms is important in preventing postharvest decay, we should investigate the characteristics of the blueberry surface microbial communities and provide changes for microbial communities to improve preservation methods. Temperature changes have a great impact on the control of bacterial succession on the surface of fruits, such as sweet cherries [8], avocados [9], and fresh-cut dragon fruits [10]; thus, the changes in microflora or specific microorganisms during storage were studied. However, the changes and patterns of microorganisms of blueberries during the postharvest storage period have not been studied. In this study, the succession characteristics and changes in blueberry flora under normal-temperature and low-temperature environments were explored, which laid a foundation for the later creation of a targeted degradation inhibition and preservation technology.

High-throughput sequencing can efficiently and conveniently obtain microbial community diversity information, and accurately grasp the diversity and richness of microorganisms. It is commonly applied to rhizosphere microbial communities, especially for the study of soil and plant inter-root microorganisms [11]. The relationship between microbial changes on the surface of fruits and vegetables, and quality is only partially explored [12]. Zhang et al. [8] studied the changes in cherry surface microorganisms and distinguished differences in the surface microorganisms of rotten and unrotten cherries, but the changes in the cherries’ quality were not investigated. The relationships between the changes in microbial communities on the surface of *Flammulina velutipes* and the taste under different packaging conditions have been reported [13]. The microbial changes on the surface of fruits and vegetables after harvesting have an important relationship with their quality, and we aim to reveal the microbial status of the blueberry surfaces after storage at different temperatures by high-throughput technology, and to provide a microbial basis for blueberry postharvest preservation.

Cold storage is a reliable way to maintain the postharvest quality of blueberries. The actual postharvest storage temperature of the blueberries fluctuates greatly, which leads to the deterioration of the blueberries and affects their edible quality [14]. Fruits are usually stored at different temperatures to maintain their quality after harvest [15,16,17]. In commercial sales activities, low temperature (4 °C) and room temperature (25 °C) are the two most commonly used storage temperatures [18]. Therefore, combined with the postharvest storage of fruits, the effect of storage temperature on the surface microorganisms of fruits was studied.

In this study, high-throughput sequencing technology was used to analyze the bacterial and fungal communities on the surface of the blueberry fruits preserved at 25 °C and 4 °C. The symbiotic relationships between various microbial groups and their relationships with the storage state were determined, and biomarkers of microorganisms during storage were identified. Studying the relationship between the changes in microbial communities on the surface of blueberry fruits under different storage conditions, which is based on preservation quality, is helpful to develop more accurate preservation and control methods.

## 2. Materials and Methods

### 2.1. Sample Preparation

The blueberry fruits of the cultivar ‘Baldwin’ were harvested at Baima Town, Lishui District, Nanjing city, Jiangsu Province, China (31°53′ N, 119°16′ E). The sample collection was based on Shi et al.’s method with slight modifications [19]. An amount of 10 kg of fresh blueberries with the same maturity and uniform size were harvested by hands wearing sterile gloves, placed in a plastic box, and transported to the laboratory within 2 h. All of the blueberry samples were pre-cooled at 4 °C for 12 h in the dark, then were divided into 2 groups, and collected into a PET box (138 × 85 × 35 mm) with the lid closed of appropriately 200 g per box and 10 boxes per group. The 2 groups were stored at 25 °C (T_25_) and 4 °C (T_4_) for 21 days, respectively. The humidity of both groups was 85%. The fresh blueberries (unstored) were marked as original samples (CK).

### 2.2. Preservation Quality Indices

The L*, a* and b*, firmness, and decay rates were measured using the methods described by C. Mannozzi et al. [20] with appropriate modifications.

The surface color changes of blueberry fruits during storage were measured using a 3 nh colorimeter (3NH SR-66; Shenzhen 3NH Technology Co., Ltd., Shenzhen, China). Three parameters, L*, a* and b*, were measured for each blueberry sample. Five blueberry fruits were measured in each box, and the same blueberry fruit was measured three times in different positions. The mean value was calculated as the experimental data.

The firmness of the blueberry fruit was measured by TPA using a fruit firmness tester with a 2 mm stainless steel probe (GY-4; Jinkelida Instrument Co., Ltd., Beijing, China), and expressed as N. Five blueberry fruits were measured in each box, and each fruit was measured twice in different parts. The result was the average value of the firmness of 5 blueberry fruits. The decay rate was the ratio of the number of rotten blueberry fruits to the total number of blueberry fruits in each box.

### 2.3. Microbiological Analyses

#### 2.3.1. Microbial Collection on Blueberry Surfaces

The blueberry fruits were stored at different temperatures for 21 days, and then 10 fruits were randomly selected from each treatment. The sample collection procedure required by Dikio Gene Sequencing Biologics (Genedenovo Co., Ltd., Guangzhou, China) was performed as follows. The entire surface of the 10 blueberry fruits was scrubbed with a sterile cotton swab to collect the microorganisms, and the integrity of the blueberry skin was maintained. Then, the sterile cotton swabs were packaged in sterile centrifuge tubes and stored in a −80 °C refrigerator (DW HL528S, Zhongke Meiling Co., Ltd., Hefei, China), and each experimental treatment was repeated three times.

#### 2.3.2. DNA Extraction, High-Throughput Sequencing, and Data Control

A HiPure Soil DNA Kit (D3142, Guangzhou American Biotechnology Co., Ltd., Guangzhou, China) was used to extract the total deoxyribonucleic acid from the blueberry surface microorganisms and test the integrity of the nucleic acid samples. Each sample was repeated three times. The full-length DNA of 16S and ITS was amplified by PCR using the V3-V4 hypervariable region of the bacterial 16S rRNA (341F: CCTACGGGNGGCWGCAG, 806R: GGACTACHVGGGTATCTAAT) and the ITS2 region of fungi (ITS3 _ KYO2: GATGAAGAACGYAGYRAA; iTS4: TCCTCCGCTTATTGATGC) as primers. Purified amplicons were pooled in equimolar amounts and paired-end sequenced (PE250) on an Illumina platform (Illumina, San Diego, CA, USA) according to standard protocols. The Illumina raw sequence reads were deposited into the NCBI Sequence Read Archive (SRA) under accession number PRJNA935852.

Raw reads were further filtered to obtain high-quality clean reads using FASTP (version 0.18.0). The clean tags were clustered into operational taxonomic units (OTUs) of  ≥ 97% similarity using the UPARSE (version 9.2.64) pipeline. The representative OTU sequences were classified into organisms by a naive Bayesian model using the RDP classifier (version 2.2) based on the SILVA database (version 132) and ITS2 database (version update_2015), with a confidence threshold value of 0.8.

### 2.4. Bioinformatics Analysis

Alpha diversity analyses of Chao1, ACE, Shannon, Simpson, Good’s coverage and Pielou’s evenness index were calculated in QIIME [21] (version 1.9.1). Beta diversity analyses of multivariate statistical techniques, including PCoA (principal coordinates analysis) based on (un)weighted UniFrac distances, were generated in R by the ‘vegan’ package (version 2.5.3). Statistical analyses of Tukey’s HSD test, and the Adonis and ANOSIM tests were calculated in R by the ‘vegan’ package (version 2.5.3). The stacked bar plot of the community composition, heatmap of species abundance, between-group Venn analysis, biomarker features, and Mantel test were completed, and the Pearson correlation coefficients between environmental factors and species were plotted in R software Version 2.2.2 [22].

Analyses of variance (one-way ANOVA) were performed by using the SPSS System, and mean separations were compared by using Duncan’s test. Differences were considered significant at *p* < 0.05.

## 3. Results

### 3.1. Storage Temperature Affects Microbial α-Diversity on Blueberry Surfaces

The α-diversity indices of the bacterial and fungal communities were calculated based on the rarefied OTU profiles. The indices of each experimental group are shown in Table 1, with a total of 363,253 and 294,645 row sequences of bacteria and fungi, respectively. After quality control, there were approximately 350,883 and 279,157 effective bacterial and fungal sequence reads, respectively. In total, there were 1, 178 and 987 OTUs in the bacterial and fungal communities. The number of bacterial OTUs was lower in CK than in T_25_ and T_4_ (*p* < 0.05), and the highest number of OTUs was found in T_4_. There was a significant difference (*p* < 0.05) in the fungal abundance between T_25_ and T_4_; there were fewer fungal species present in T_25_ than in T_4_.

The statistically significant differences between the two treatment groups are shown in Table 2. The α-diversity index in T_25_ was significantly lower than that in CK. This trend was significantly higher in fungi than in bacteria (*p* < 0.05). The α-diversity index in CK and T_4_ had no significant change with only a marginal difference in both bacteria and fungi. Significant differences in the α-diversity indices between T_4_ and T_25_ were observed on the blueberry surface for all bacteria and fungi. Our results showed that low temperature (4 °C) significantly maintained the α-diversity of the microbiota on the blueberry surface compared with high temperature (25 °C). Moreover, on day 21, the α-diversity of the bacterial community was more sensitive at 4 °C, and the α-diversity of the fungal community was more sensitive at 25 °C. Differential analysis was performed using the ratio of 16S to ITS Shannon and Simpson diversity indices, the results were shown in Appendix A, and the ratio was used to reflect the balance of bacterial and fungal diversity. A lower ratio indicates that the abundance of bacteria was similar to that of fungi, and the diversity of bacteria and fungi within the sample was more balanced. A significant difference in the ratio between subgroups indicates a significant difference in species balance, which indicates that our results are accurate and reliable.

### 3.2. Storage Temperature Affects Microbial Community Composition and Changes on Blueberry Surfaces

Further analysis of the differences in microbial communities on the surface of the blueberry fruits under different temperatures based on β-diversity was performed. Figure 1 showed that the bacterial and fungal community compositions were significantly different with CK. This result indicated that storage at different temperatures changed the microorganisms on the surface of blueberries, which was consistent with the α-diversity results.

The PCoA plots for bacteria and fungi are shown in Figure 1, and the results show the differences and similarities between bacterial and fungal communities for all groups, with principal component 1 (PCo1) and principal component 2 (PCo2) shown as the two eigenvalues that led to the greatest differences between samples, with contributions of 63.15% and 18.99%, respectively, and similarity r = 0.943 (*p* = 0.001). The similarity between bacterial and fungal communities in CK samples and T_4_ samples was higher under storage at 4 °C for 21 days, but the similarity between bacterial and fungal communities in CK samples and T_25_ samples was minimal under storage at 25 °C for 21 days.

The box plot of the microbial community distances on the surface of the blueberry fruits was based on Bray–Curtis distances at the OTU and other classification levels. The mean values of sample distances within (within) and between (between) groups for each comparison group were determined based on Welch’s *t*-test, and the response patterns of bacteria and fungi to the storage environment were directly compared based on the graph. In Appendix A, the bacterial and fungal microbial communities responded to the storage temperature with the same pattern, and there was a highly significant difference between treatment groups. The yellow dots in Appendix A indicate a pair of samples, and the horizontal/vertical axes indicate the distance between pairs of samples in ITS sequencing and 16S sequencing data, respectively.

The relative abundances of bacteria and fungi above 0.1% at the phylum and genus levels are shown in Appendix A. A total of 16 bacterial phyla and 92 genera were detected; meanwhile, 4 fungal phyla and 59 genera were detected on the blueberry surface. Figure 2A showed the top 10 bacterial phyla in terms of abundance, such as Proteobacteria, Cyanobacteria, Actinobacteria, Bacteroidetes, and Firmicutes. The dominant phyla in fresh blueberries (CK) were Proteobacteria (37.15%) and Cyanobacteria (50.59%), and after storage, the major phyla accounted for 89.96% and 64.13% of Proteobacteria in T_25_ and T_4_, respectively. Similarly, the phylum Proteobacteria significantly increased at the end of storage, but the phylum Cyanobacteria significantly decreased after 21 days of storage. Ascomycota, Basidiomycota, and Anthophyta were the three most abundant fungal phyla, accounting for approximately 72.93%, 83.54% and 58.18% in the CK, T_25_ and T_4_ groups, respectively. After storage, the Ascomycota and Basidiomycota phyla showed an increase in abundance at 25 °C, while their abundance decreased at 4 °C. The abundance of the Anthophyta phylum decreased significantly after storage at different temperatures.

Figure 2B showed that there were four groups of bacteria on the surface of fresh blueberries with high abundance, *Methylobacterium* (7.84%), 1174-901-12 (7.35%), *Sphingomonas* (2.24%), and *Burkholderia-Caballeronia-Paraburkholderia* (5.04%); however, when stored for 21 days at 25 °C, the abundance of the four bacterial groups decreased significantly. *Gluconobacter* increased significantly during storage at 25 °C but not at 4 °C. At the genus level, the most abundant fungi were *Zasmidium*, *Talaromyces*, *Aspergillus*, *Uwebraunia*, *Alternaria*, *Strelitziana*, *Colletotrichum*, and *Aureobasidium*. Two fungal genera, *Uwebraunia* and *Aureobasidium,* were detected, accounting for approximately 27.27%, and 5.60%, respectively, in CK, but the abundances of these fungal genera decreased significantly after storage at 4 °C. The abundances of *Zasmidium* (approximately 12.07%), *Talaromyces* (approximately 29.06%), and *Aspergillus* (approximately 25.84%) were the highest after storage at 25 °C.

The microbial communities on the surface of the blueberry fruit interacted with each other during storage, so the co-occurrence patterns of individual bacterial and fungal genera can be quantified through the dataset. Figure 3 showed correlations between most bacteria and fungi on the surface of blueberry fruits after being stored at different temperatures for 21 days. *Proteobacteria* had a high positive correlation with most of the fungi, while *Firmicutes* had a high positive correlation only with Ascomycota and *Chlorophyta. Acidobacteria* synergized with most of the fungal phyla, but *Actinobacteria* had both synergistic and antagonistic effects on the fungi. *Bacteroidetes* and *Planctomycetes* had positive correlations with fungi, and Basidiomycota had a high positive correlation with *Actinobacteria*.

### 3.3. Major Members of the Microbiota on Blueberry Surfaces

The dominant microbial species should be considered when studying the changes in the microbial communities on the surface of blueberry samples at different storage temperatures. According to the number of genus levels in Figure 4A,B, 36.11% (91/252) bacterial and 43% (83/193) fungal OTUs were shared among all three groups, indicating that there were significant differences. In Figure 4C,D, bacteria belonging to *Thermobrachium*, *Burkholderia-Caballeronia-Paraburkholderia*, *Methylobacterium,* and *Sphingomonas* were indicator microorganisms in CK. *Gluconobacter* was the indicator species in T_25_; there were more indicator microorganisms in T_4_ than in the other groups, and they had relatively high abundances, e.g., *Bdellovibrio*, *Tepidisphaera*, *Tatumella*, *Methylocella*, *Jatrophihabitans*, *Cystobacter*, *Mucilaginibacter*, *Terriglobus, Singulisphaera*, *Geodematophilus*, *Novosphingobium*, and *Tepidisphaera*. Fungi indicator genera in CK were *Uwebraunia*, *Aureobasidium*, *Passalora*, *Sphaerulina*, *Paraconiothyrium*, *Sporobolomyces*, *Pseudocercospora*, *Ramularia*, *Dissoconium*, *Neofusicoccum*, *Diaporthe*, and *Filobasidium*. In T_25_, the indicator genera were *Aspergillus*, *Talaromyces*, *Colletotrichum*, and *Schizophyllum*. In T_4_, the indicator genera were *Phaeophleospora*, *Ascochyta*, *Papiliotrema*, *Dioszegia*, *Neocladophialophora*, *Keissleriella*, and *Stagonospora* species.

Regarding different storage temperatures, the LDA (value higher than 4.0) effect size (LEfSe) of bacteria and fungi was analyzed in Figure 5. There were twenty-two taxa of different bacterial markers and twenty-four taxa of different fungal markers in those three treatment groups. We observed that nine taxa of bacteria were enriched in CK, consisting of nine genera (*oxyphotobacteria, cyanobacteria*, etc.). The T_25_ group was enriched in five taxa, including *Acetobacterales, Alphaproteobacteria*, and *Proteobacteria*, and the T_4_ group was enriched in *Beijerinckiaceae* and *Rhizobiales*. A total of twenty-four taxa dominated the fungal microorganisms, consisting of fourteen taxa such as *Dothideomycetes, Capnodiales*, *Dissoconiaceae*, and *Uwebraunia* in the CK group, five genera such as *Pleosporales* and *Alternaria* in the T_4_ group, and *Talaromyces*, *Trichocomaceae* and *Saccharomycetales* in the T_25_ group. Overall, the bacterial and fungal community compositions on the surface of blueberry fruits were different under the two storage temperatures.

### 3.4. Function Prediction and Correlation between Blueberry Quality and the Microbial Community during Storage

Table 3 lists the changes in the sensory quality of the blueberry fruits during the storage period. On day 21, the decay rate of fruits stored at 25 °C was 56.33%, which was significantly higher than that at 4 °C. Similarly, the weight loss rate of blueberries treated at 4 °C was significantly lower than that of blueberries treated at 25 °C. The firmness of blueberries stored at 4 °C was 155.73 ± 29.06 on day 21, which was twice as high as that at 25 °C.

L* is between zero for white and one-hundred for black, representing brightness, and a* and b* represent the green–red value and blue–yellow value, respectively. During storage, L* in all blueberry samples at different temperatures was not significantly different. a* decreased in the two treatment groups on day 21 but was significantly higher in the 4 °C treatment group than in the 25 °C group. At 21 days after storage, the b* value of blueberries stored at the two temperatures was significantly higher than that of CK and was lower in the 4 °C treatment group than that in the 25 °C group (*p* < 0.05).

Figure 6 showed that the bacterial and fungal OTUs and their Shannon indices were correlated with environmental factors, and the variability of the main environmental factors was analyzed by a partial Mantel test. Overall, the five preservation quality indices had appreciable effects on the bacterial and fungal community compositions, but significantly weaker effects on the bacterial community alpha diversity compared with the fungal community. The 16S_Shannon bacterial diversity composition was correlated with the environmental factor Color_a (R = 0.07185, *p* = 0.235); the Color_b correlation (R = 0.3257, *p* = 0.059) was weaker than the ITS_Shannon fungal diversity Color_a correlation (R = 0.4342, *p* = 0.018) and Color_b correlation (Mantel’s R = 0.7441, *p* = 0.003).

The gene function prediction of the microbial communities was based on the PICRUSt2 database with KEGG functional classification analysis. The results showed that the bacterial genes’ function prediction on the surface of the blueberry samples was mainly associated with four categories: metabolism, genetic information processing, cellular processes, and environmental information processing (Figure 7). Approximately one-half of the genes were related to metabolism, and the largest share was carbohydrate metabolism, amino acid metabolism, and cofactor and vitamin metabolism, followed by microbial biodegradation and metabolism, terpene and polyketide metabolism, lipid metabolism, and energy metabolism.

## 4. Discussion

### 4.1. Microbial Community Diversity and Composition at Different Storage Temperatures

The quality status of fruit during the postharvest storage is closely related to the microorganisms on the fruit’s surface [23]. Harvested fruits are threatened by pathogens, which cause quality losses of up to 55%; for example, losses resulting from anthracnose pre-harvest are typically 3% to 5%, and postharvest losses can reach 100% [24]. The surface of fruit is colonized by complex microbial communities that are often resilient [25], but there are relatively few reports on the fruit surface microbes. The surface microbial community is a complex functional network of bacteria and fungi on the plant surface, including symbiotic interactions between microorganisms and visible interactions with the host plants [26]. In the past few years, significant progress has been made in understanding the role of bacterial and fungal communities as part of the whole organism [27]. It is well known that fruits are susceptible to decay after harvest, resulting in loss of commercial value, and one of the reasons for this is the infection of microorganisms and pathogens. Therefore, it is crucial to investigate the changes in the microbial communities on the surface of fruits before and after storage for the accurate development of preservation methods.

The blueberries were stored at 25 °C for 21 days, and some fruits rotted (Table 1). In this case, the α-diversity of bacteria and fungi changed differently. According to the various α-indices, at 25 °C, storage had little effect on the bacterial diversity of the blueberry surface, but it significantly reduced the fungal diversity, indicating that the change in the fungal diversity on the surface of some decayed blueberries was more sensitive than that of the bacteria. Similar conclusions were found in a study of cherry surface microorganisms. In contrast to the 25 °C, the fruits at 4 °C showed the opposite pattern: the 4 °C low-temperature environment significantly maintained the α-diversity of bacteria on the surface of the blueberry samples and had little effect on the α-diversity of fungi, indicating that the bacterial community diversity obviously responded to low temperature compared with fungi.

### 4.2. Microbial Community Biomarker Dynamics at Different Storage Temperatures

Under different temperatures, the bacterial biomarker vs. fungal biomarker numbers may vary, and high temperatures may increase the numbers. Notably, at 25 °C, some taxa were significantly more abundant than those at 4 °C, which may affect the quality of the fruit. On the other hand, some spoilage bacteria also existed on the surface of the blueberries, such as *Aspergillus*, *Talaromyces*, *Aureobasidium*, *Colletotrichum*, *Schizophyllum*, *Sphaerulina*, and *Neofusicoccum* [28] (Figure 4C,D), which are potential causes of fruit mildew or decay. These bacteria and fungi mainly originate from soil, atmospheric, and water sources. *Aspergillus*, *Talaromyces*, *Colletotrichum,* and *Schizophyllum* belong to the phylum Pathomycete, and these microorganisms increased in abundance during storage at 25 °C. *Aspergillus*, *Talaromyces*, *Colletotrichum,* and *Schizophyllum* and other pathogenic bacteria are common pathogens on many fruits and usually cause different diseases of fruits, such as gray mold and anthracnose [29,30]. This study focused on the interaction between different microorganisms on the surface of blueberries to distinguish interactions between potential pathogenic microorganisms [6]. Mamphogoro et al. [31] revealed the close relationship between the microorganisms and postharvest losses. According to previous studies, fruit rot mildew is caused by certain microorganisms, and synergistic or antagonistic effects were also observed in the process [32]. Elisa Cabrera Díaz et al. [9] studied the survival ability of pathogens on the surface of stored avocados, showing that if the initial cell number was large enough, the pathogens existed longer and they were more reproductive. In this study, Figure 2 and Figure 4 showed similar results. In order to meet the requirements of blueberry consumption and storage quality, it is important to keep the microbial level on the surface of blueberries low. Otherwise, high microbial levels on the surface of blueberries will pose a risk to consumers for ready-to-eat foods.

### 4.3. Microorganism-Blueberry Fruit Quality Interrelationships

Temperature is an important factor that affects the postharvest quality of fruits. Normal-temperature storage and low-temperature storage are both important postharvest storage methods for fruits [33,34]. Therefore, it is of great significance to study the storage mechanism and develop a storage technology. The microbial communities on the surface of blueberries stored at different temperatures changed significantly compared with the original samples (CK) (Figure 2). Some researchers have verified that changes or disorders in the composition of microbial flora lead to outbreaks of plant diseases [25,27]. Compared with the original samples (CK), the α-diversity of microorganisms on the surface of blueberries at the 4 °C low-temperature storage was higher. High levels of microbial diversity can maintain the balance of microorganisms, thereby maintaining the relative stability of the microbial communities and plant states [26]. In Figure 3, the microorganisms have mutual regulatory relationships. Decay mildew is the result of the vigorous and dominant growth of several microorganisms on the fruit surface, not the role of a single microbial member [35]. We have verified that low temperature is a reliable means to delay blueberry decay.

The appearance quality of stored blueberries and important indicators can intuitively reflect the state of change of the stored blueberries. With the extension of the blueberry storage time at low temperature and room temperature, shown in Table 3, the a* value increased, indicating that the blueberry sample changed from green to red; the b* value increased significantly, indicating that the room temperature and low temperature accelerated the color change of the fruit, which is consistent with the color change dynamics of blueberry fruit during storage. Hardness is a direct indicator used to evaluate the fruits’ quality, and the change in hardness during storage may be related to microbial action [36]. Our results showed that at the 25 °C storage, the hardness of blueberries decreased, and the softening of blueberries accelerated. A low temperature can delay softening of blueberry fruits during storage, which is beneficial to the maintenance of the storage quality. However, the higher temperature increased the respiration rate and microbial activity, leading to softening [37]. The surface microbial diversity of blueberry fruit was inhibited by storage at 4 °C (Table 2 and Figure 5), thereby reducing the decay rate of the blueberry fruits (Table 3). Additionally, our study showed that some bacteria affected the regulation of sugar metabolism, vitamin metabolism, and metabolites in blueberries based on a functional prediction of KEGG pathway analysis (Table 3). Therefore, the change in blueberry quality during storage was closely related to its surface microorganisms, which is consistent with a previous study showing that the community structure changed significantly with changes in parameters [38]. The change in the bacterial community was consistent with the change in the blueberry quality.

Collectively, fruit mildew and decay have a higher correlation with various pathogenic fungi [39,40]. However, the changes in the fruit microbial communities before and after storage at 25 °C, and the interaction between microbial changes and host environmental factors at different storage temperatures have rarely been explored. Our investigation provided some evidence. First, the storage process changed the composition of bacterial or fungal communities in the samples (Figure 2). Second, low-temperature storage maintained more surface diversity than room-temperature storage (Table 1 and Table 2). Again, the proportion of fungal microbes was greater than that of bacterial microbes and was associated with color changes and weight loss rates of the blueberries (Figure 7). In addition, functional analysis showed that fruit metabolism provided microbial eutrophic habitats and increased the number of microorganisms, resulting in fruit decay (Figure 6). The microbial community plays an important role in postharvest mildew and quality changes in fruits [41,42]. Few studies have compared changes in bacterial and fungal communities with quality degradation. This study preliminarily showed that the change in the bacterial community was more obvious than that in the fungal community at low temperatures, and the change in the fungal community was dominant at room temperatures. This may be related to microbial community function, but further research is still needed to further explain the mechanism. We consider our future research would start from two aspects: first, the development of precise sterilization and preservation methods for biological indicators of microorganisms on the surface of blueberries, so that the level of microorganisms on the surface of blueberries may be controlled within an acceptable range; second, the correlation between various physicochemical indexes of blueberries such as soluble sugars, titratable acids, anthocyanins, and surface fruit waxes, and microbial community changes should be studied.

## 5. Conclusions

In this study, the structure of microbial communities on the surface of blueberries stored at different temperatures was investigated, and the storage quality of blueberry fruits was determined. The bacterial biodiversity was significantly higher than that of the fungi in the low-temperature storage at 4 °C, while the fungal diversity changed more significantly under storage at room temperature (25 °C). A total of 319 bacterial OTUs and 365 fungal OTUs were detected in the fresh blueberries, 364 bacterial OTUs and 236 fungal OTUs were detected in the T_25_ group, and 472 bacterial OTUs and 385 fungi OTUs were detected in the T_4_ group. Both bacterial and fungal communities showed consistent changes at the different storage temperatures. The Pearson correlation analysis showed that preservation quality indices interacted with the bacterial and fungal community compositions. The action of the blueberry surface microbial community was characterized by functional prediction associated with six categories: metabolism, genetic information processing, cellular processes, environmental information processing, organismal systems, and human diseases. This work provides a basis for the changes in the microbial communities of blueberries stored at different temperatures. However, it is still necessary to further study the mechanism of the interrelationship between blueberry quality changes and microbial communities during storage to provide a theoretical reference for the development of new storage and preservation means for fresh blueberry fruits.

## Figures and Tables

**Figure 1 foods-12-01611-f001:**
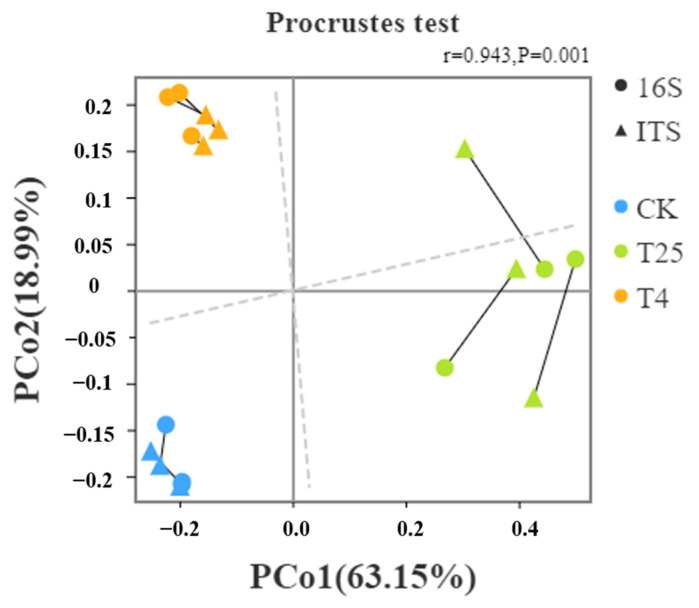
Scatter plot of 16S and ITS β-diversity index correlation ratio.

**Figure 2 foods-12-01611-f002:**
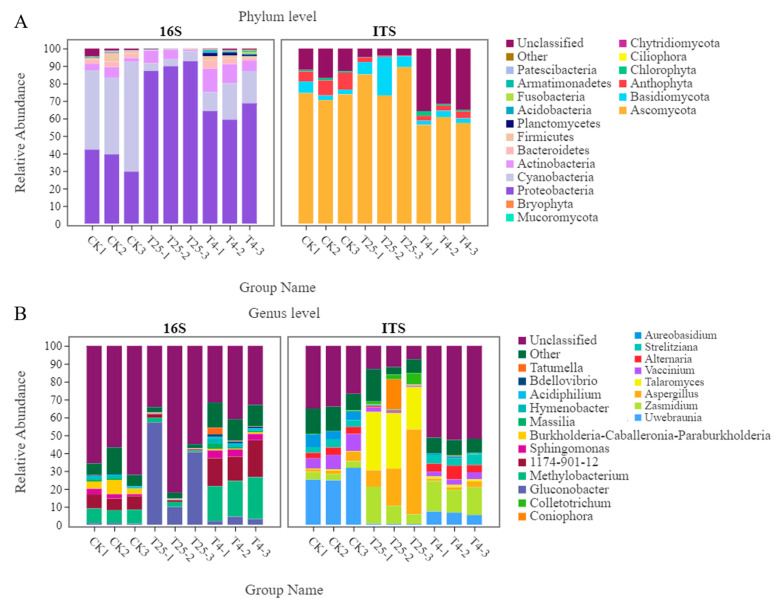
Relative abundances of bacteria and fungi at the phylum level (**A**) and genus level (**B**). CK refers to fresh blueberry samples; T_25_ and T_4_ refer to temperatures of 25 °C and 4 °C (on day 21).

**Figure 3 foods-12-01611-f003:**
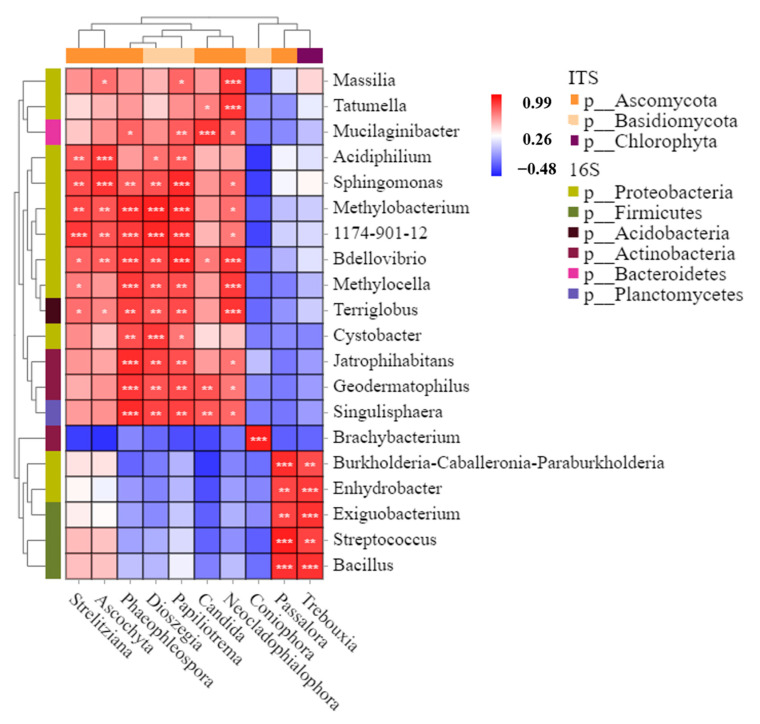
The thermograms of regulatory intensity of the correlation between bacteria and fungi on the surface of blueberry samples. Note: The similarity value was determined by calculating the Pearson correlation coefficient between paired species. Red means a positive correlation; blue means a negative correlation (* *p* < 0.05, ** *p* < 0.01, *** *p* < 0.001).

**Figure 4 foods-12-01611-f004:**
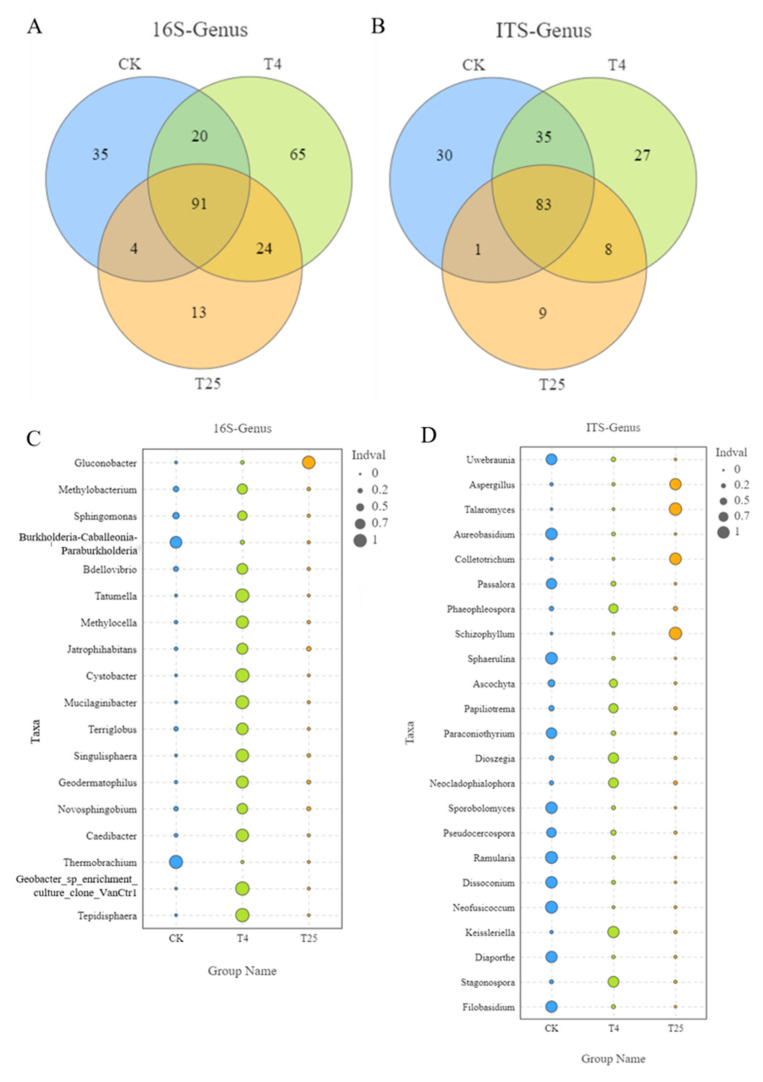
Venn plot of bacterial (**A**) and fungal (**B**) operational taxonomic units (OTUs) in all samples. Indicator diagram of bacterial (**C**) and fungal (**D**) genus level species.

**Figure 5 foods-12-01611-f005:**
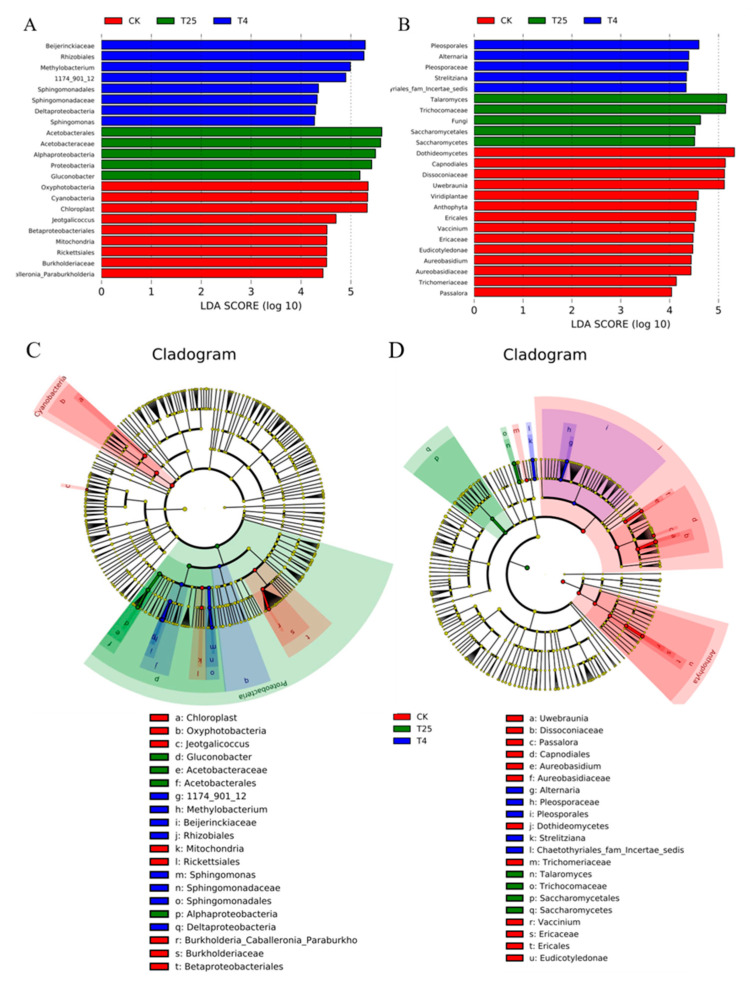
LEfSe analysis illustrating the differences in the relative abundance of taxa on the blueberry surface at different storage temperatures. The linear discriminant analysis (LDA) histogram exhibits biomarkers of fungal microbiota on the blueberry surface. The histogram shows taxa with significant differences (cut-off score ≥ 2.0). The cladogram was with significantly discriminant taxon nodes in different colors. Red and green circles mean that the taxa showed differences in relative abundance, and yellow circles mean non-significant differences; branch areas are shaded according to the highest ranked group for that taxon. (**A**,**C**): bacteria, (**B**,**D**): fungi.

**Figure 6 foods-12-01611-f006:**
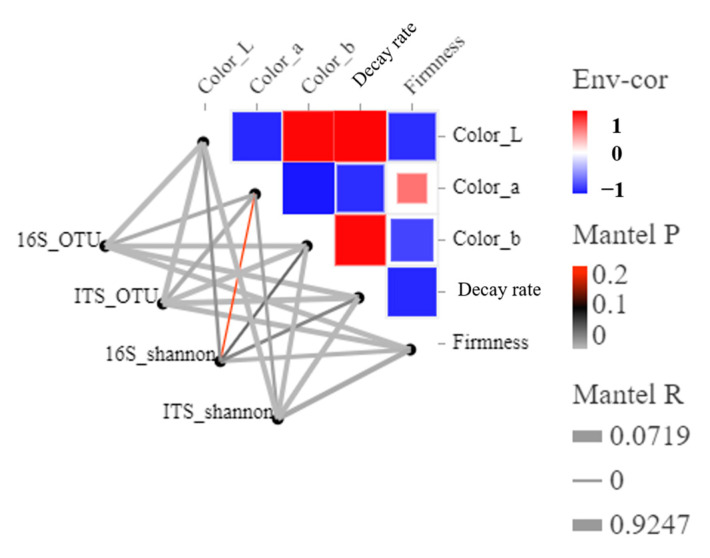
Correlations between fungal taxa (ITS) and bacterial taxa (16S) and the apparent characteristics of blueberries after storage. Red squares represent significant (*p* < 0.05 after FDR adjustment) negative correlations. Blue squares represent significant (*p* < 0.05 after FDR adjustment) positive correlations. Darker colors represent stronger correlations. Non-significant correlations are not shown.

**Figure 7 foods-12-01611-f007:**
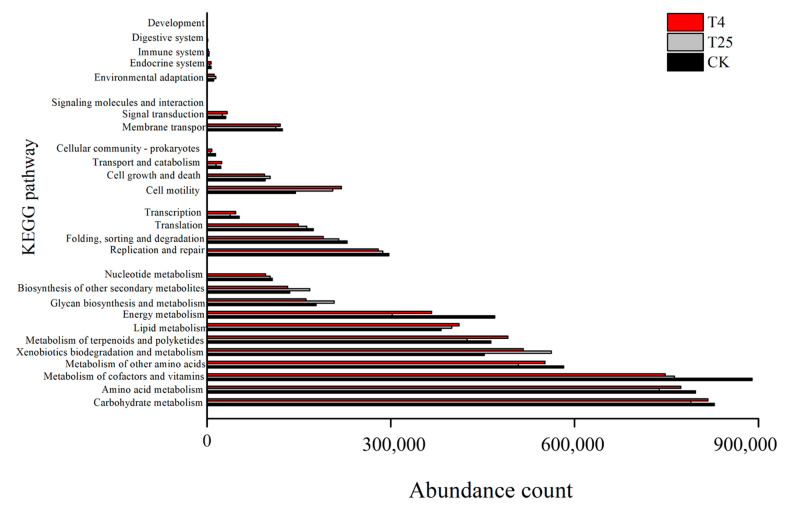
KEGG functional annotation of microbial communities on the blueberry surface.

**Table 1 foods-12-01611-t001:** Raw sequencing data of bacteria and fungi on the surface of the blueberry samples.

Storage (Days)	Group	Raw Sequences		Effective Tags		N90		OTUs		Effective Ratio (%)	
		Bacterial	Fungal	Bacterial	Fungal	Bacterial	Fungal	Bacterial	Fungal	Bacterial	Fungal
0	CK	126,481 ± 2018 ^a^	109,728 ± 3229 ^a^	122,473 ± 1798 ^a^	106,061 ± 3246 ^a^	441	319 ± 1	342 ± 11 ^b^	365 ± 16 ^a^	96.84 ± 0.14 ^a^	96.65 ± 0.17 ^a^
21	T25	130,028 ± 3704 ^a^	89,113 ± 483 ^b^	125,765 ± 3842 ^a^	82,275 ± 396 ^c^	441	326 ± 4	364 ± 8 ^b^	236 ± 13 ^b^	96.71 ± 0.29 ^a^	92.34 ± 0.91 ^b^
	T4	106,744 ± 3293 ^b^	95,803 ± 2514 ^b^	102,646 ± 3010 ^b^	90,820 ± 2629 ^b^	441	317 ± 0	472 ± 24 ^a^	386 ± 17 ^a^	96.17 ± 0.22 ^a^	94.79 ± 0.31 ^a^

Note: Means with standard deviation are shown. Different lowercase letters (a, b, c) in the same column indicate significant difference (*p* < 0.05) of means between groups.

**Table 2 foods-12-01611-t002:** Significance analysis of bacterial and fungal α−diversity indexes.

Group	Shannon		Simpson		Chao1		Sobs		Good’s Coverage	
	Bacterial	Fungal	Bacterial	Fungal	Bacterial	Fungal	Bacterial	Fungal	Bacterial	Fungal
CK	3.85 ± 0.49 ^b^	4.57 ± 0.16 ^b^	0.76 ± 0.064 ^b^	0.9 ± 0.013 ^a b^	425.77 ± 20.93 ^b^	414.24 ± 14.9 ^b^	341.67 ± 11.35 ^b^	365 ± 16.44 ^b^	0.999	0.999
T25	2.64 ± 0.26 ^c^	3.63 ± 0.097 ^a^	0.69 ± 0.034 ^b^	0.86 ± 0.003 ^b^	453.06 ± 10.68 ^b^	291.8 ± 21.09 ^a^	363.67 ± 8.455 ^b^	236 ± 12.77 ^a^	0.999	0.999
T4	5.22 ± 0.2 ^a^	4.47 ± 0.046 ^b^	0.92 ± 0.013 ^a^	0.91 ± 0.004 ^a^	528.33 ± 4.84 ^b^	434.78 ± 16.86 ^b^	472.33 ± 23.95 ^a^	386 ± 17.01 ^b^	0.999	0.999
CK vs. T25	0.12	0.011 *	0.37	0.08	0.33	0.012 *	0.2	0.0042 **	0.21	0.036 *
CK vs. T4	0.092	0.6	0.12	0.43	0.034 *	0.41	0.018 *	0.42	0.31	0.17
T4 vs. T25	0.0017 ^**^	0.0051 **	0.011 *	0.0007 **	0.0095 **	0.007 **	0.034 *	0.0028 **	0.15	0.89
CK vs. T4 vs. T25	0.0042 **	0.0021 **	0.017 *	0.0082 **	0.0047 **	0.0031 **	0.0029 **	0.0012 **	0.13	0.14

Note: A two-sided *t*-test was used to compare changes in the α-diversity index among the three groups. ** p <* 0.05, *** p* < 0.01. A three-sided Tukey-HSD was used to compare changes in the α-diversity index among the three groups. ** p* < 0.05, *** p* < 0.01. Different lowercase letters (a, b, c) in the same column indicate significant difference (*p* < 0.05) of means between groups. Shannon and Simpson comprehensively reflect the richness and evenness of species. The more uniform the distribution of species in the sample, the higher the diversity. The ACE, Chao, and sobs indices show the species richness of the samples. Peilou reflects the uniformity, the larger the value, the more uniform. The PD index is based on the phylogenetic characteristics of the OTU sequence evolution tree to assess the degree of diversity, that is, lineage diversity. Good’s Coverage reflects the sequencing saturation of the sample.

**Table 3 foods-12-01611-t003:** Preservation quality indices of blueberry samples at different storage temperatures.

Storage (Days)	Group Name	Firmness (N)	Decay Rate (%)	L*	a*	b*
0	CK	137.27 ± 39.91	13.64 ± 0.53	26.42 ± 1.50	−3.12 ± 1.40	−1.56 ± 0.90
21	T25	76.20 ± 2.73 ^b^	9.31 ± 0.67 ^b^	27.62 ± 3.317 ^a^	7.70 ± 3.10 ^a^	3.82 ± 3.79 ^a^
	T4	155.73 ± 29.06 ^a^	12.33 ± 1.01 ^b^	26.76 ± 2.675 ^a^	4.20 ± 3.42 ^b^	0.76 ± 3.31 ^b^

Note: Each value represents the mean of six replicates ± standard devia. Means with standard deviation are shown. Different lowercase letters (a, b) in the same column indicate significant difference (*p* < 0.05) of means between groups.

## Data Availability

The data presented in the study are deposited in the NCBI Sequence Read Archive (SRA) repository, accession number PRJNA935852.

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
