# Peer review of "The Impact of Storage Temperature on the Development of Microbial Communities on the Surface of Blueberry Fruit"

_foods, 2023, doi:10.3390/foods12081611_

Round 1
Reviewer 1 Report
line 72 - In my opinion, the term "symbiotic relationship" is not appropriate. Check through the entire text
line 82 - Explain in more detail what "harvested manually" means - with bare hands, gloves, with some sanitation
Author Response
Dear Editor and Reviewers:
Thank you for your letter and for the reviewers’ comments concerning our manuscript entitled“The Impact of Storage Temperature on the Development of Microbial Communities on the Surface of Blueberry fruit”(Manuscript ID: foods-2261235). Those comments are all valuable and helpful for revising and improving our paper, as well as the important guiding significance to our researches. We have studied comments carefully and have made revision which we hope meet with approval. Responses to reviewer comments are marked in red in the attached document. The main corrections and revisions in the paper can be found in the revised manuscript.

Reviewer 2 Report
Food waste is one of the current main worries and big efforts are being done in order to tackle it. Between different foodstuffs, fruits are one of the most perishable, mainly due to their high water and sugar content, and the presence of natural microbiota that can evolve by consuming fruit nutrients and spoiling the fruit. Several preservation techniques are nowadays being used. Nevertheless, one of the most widespread is cold storage. The study of the influence of temperature storage over natural flora present in fruit could strongly help the optimization of storage conditions, even more with such a comprehensive study, including bacterial and fungal communities, and the vanguard techniques used. Nevertheless, here you have some suggestions or little points to clarify and even improve the study.
Author Response

(The authors gave the same response as above.)

Reviewer 3 Report
This is a very interesting manuscript dealing with the current topic of microbial diversity and its influence on the behavior of fruits stored in different environmental conditions (i.e. temperature). The topic addressed in the manuscript is original and interesting for the scientific community, and the techniques used in the work are novel.
Some minor aspects are recommended to be analyzed, as well as the opportunity to be taken into consideration by authors:
-overall, the comparison of the behavior of the microbial communities for blueberry fruits is performed in respect to sweet cherries, however there in no clear justification for this approach. It is recommended to make a better connection with this aspect, which is then discussed in the rest of the manuscript.
-it is not very clearly highlighted which are the five environmental factors that were measured in this study;
-please specify more clearly if the study addresses the plant or the human pathogenic microorganisms
-line 376 – please verify the accuracy of the verb “to curb” associated with microorganisms / microbes
-line 386 – please verify it the term researchers is not more appropriate that scholars
Author Response

(The authors gave the same response as above.)

Reviewer 4 Report
In the paper with the title The Impact of Storage Temperature on the Development of Microbial Communities on the Surface of Blueberry fruit, high-throughput sequencing technology was used to analyze the bacterial and fungal communities on the surface of blueberries stored at 25°C and 4°C. The symbiotic relationship between different microbial groups and their relation to storage condition were determined and biomarkers of microorganisms during storage were identified. For the development of more accurate preservation and control methods, studying the relationship between changes in microbial communities on the surface of blueberries under storage conditions is useful.
I recommend the authors to specify in chapter 2, more precisely to subchapters 2.1., 2.2. and 2.3.1. if the methods used are original or if they are taken from specialist articles.
Author Response

(The authors gave the same response as above.)
